# When ELIZA meets therapists: A Turing test for the heart and mind

S. Gabe Hatch[1], Zachary T. Goodman[1,2], Laura Vowels[3], H. Dorian Hatch[1,4]*, Alyssa L. Brown[5], Shayna Guttman[6], Yunying Le[7], Benjamin Bailey[8], Russell J. Bailey[8], Charlotte R. Esplin[9], Steven M. Harris[10], D. Payton Holt, Jr.[11], Merranda McLaughlin[12], Patrick O'Connell[13], Karen Rothman[14], Lane Ritchie[7], D. Nicholas Top, Jr.[8], Scott R. Braithwaite[5]

1 Hatch Data and Mental Health, Orem, Utah, United States of America, 2 Department of Psychiatry, University of California San Diego, San Diego, California, United States of America, 3 Department of Psychology, University of Lausanne, Lausanne, Switzerland, 4 Department of Psychology, The Ohio State University, Columbus, Ohio, United States of America, 5 Department of Psychology, Brigham Young University, Provo, Utah, United States of America, 6 Department of Psychology, University of Miami, Coral Gables, Florida, United States of America, 7 Department of Psychology, University of Denver, Denver, Colorado, United States of America, 8 Student Health Services, Utah Valley University, Orem, Utah, United States of America, 9 Department of Psychology, Sam Houston State University, Huntsville, Texas, United States of America, 10 Department of Family Social Science, University of Minnesota Twin Cities, Minneapolis, Minnesota, United States of America, 11 The Marriage and Family Clinic, Spanish Fork, Utah, United States of America, 12 Center of Excellence for Psychosocial and Systemic Research, Massachusetts General Hospital, Boston, Massachusetts, United States of America, 13 Department of Psychiatry and Behavioral Sciences, Emory University, Atlanta, Georgia, United States of America, 14 NYCBT, New York City, New York, United States of America

* dorian@hatchdamh.com

**Data Availability Statement:** Before data collection began, permission was sought and granted from the Brigham Young University IRB. Written consent was obtained from all participants and recruitment ran from March 8, 2024, through

## Abstract

"Can machines be therapists?" is a question receiving increased attention given the relative ease of working with generative artificial intelligence. Although recent (and decades-old) research has found that humans struggle to tell the difference between responses from machines and humans, recent findings suggest that artificial intelligence can write empathically and the generated content is rated highly by therapists and outperforms professionals. It is uncertain whether, in a preregistered competition where therapists and ChatGPT respond to therapeutic vignettes about couple therapy, a) a panel of participants can tell which responses are ChatGPT-generated and which are written by therapists (N = 13), b) the generated responses or the therapist-written responses fall more in line with key therapy principles, and c) linguistic differences between conditions are present. In a large sample (N = 830), we showed that a) participants could rarely tell the difference between responses written by ChatGPT and responses written by a therapist, b) the responses written by ChatGPT were generally rated higher in key psychotherapy principles, and c) the language patterns between ChatGPT and therapists were different. Using different measures, we then confirmed that responses written by ChatGPT were rated higher than the therapist's responses suggesting these differences may be explained by part-of-speech and response sentiment. This may be an early indication that ChatGPT has the potential to improve psychotherapeutic processes. We anticipate that this work may lead to the development of different methods of testing and creating psychotherapeutic interventions. Further, we discuss limitations

March 11, 2024. Before panel data was collected, all analyses, hypotheses, and tests were preregistered (https://osf.io/up7v4/?view_only=ef738a5211a643fa97262859f84cf33f) with the Open Science Framework. Further, data, code, and study materials can be accessed here (https://osf.io/8mnsc/?view_only=7e12583f88394f0e8db97688c0bad40f). All analyses were conducted by the second author who was blind to the condition.

**Funding:** The authors received no specific funding for this work.

**Competing interests:** The authors have declared that no competing interests exist.

(including the lack of the therapeutic context), and how continued research in this area may lead to improved efficacy of psychotherapeutic interventions allowing such interventions to be placed in the hands of individuals who need them the most.

## When AI meets couple therapy: A Turing test for the heart and mind

"Can machines think?" represents a simple question raised by Alan Turing shortly after the Second World War [1]. When proposing the "imitation game," Turing proposed an "interrogator" asking a series of questions through writing to both a human and a machine, the interrogator must decide the author of the response. Turing's original prediction was that by the end of the century (i.e., 2000), most interrogators would only guess correctly 70% of the time [1]. Less than 20 years later, ELIZA was created—one of the first chatbots capable of the imitation game [2]. ELIZA was programmed to respond as a Rogerian psychotherapist. Participants were expected to "talk" to ELIZA, and ELIZA's primary responsibility was ensuring the writer was being understood. After interacting with ELIZA, Weizenbaum noted: "ELIZA shows. . . how easy it is to create and maintain the illusion of understanding. . .. with so little machinery." [2]. Technology has vastly improved since the mid-1900s. Although some practitioners maintain that "robots [are] incapable of being in a caring relationship," [3] early and mounting evidence suggests that generative artificial intelligence (GenAI) may prove helpful in psychotherapy [2].

Several recent psychotherapy-related studies indicate the promising effects of using GenAI as an adjunct to human services or as an independent solution. For example, HAILEY, an AI-in-the-loop writing assistant has been proposed as an addition to *TalkLife*–a peer-to-peer platform for mental health support–providing just-in-time feedback [4]. Overall, 63.3% of responses incorporating feedback from HAILEY were rated equal to, or more empathic than, responses written by a human. Another adjunct solution used ChatGPT 3.5 to coach individuals in the workplace on prosocial messages of gratitude or employee recognition [5]. Respondents preferred AI-coached messages over those written by humans. Linguistic elements extracted from the messages revealed the AI-coached messages were longer, had a more positive sentiment, and used more nouns, adjectives, and verbs than the human-written messages, even after controlling for length. These written responses allow for linguistic-based (e.g., part of speech, sentiment, and word frequency) explanations of the differences found rather than relying on psychometric constructs [5].

A similar pattern of results emerges when the model is expected to generate independent content. For example, using 195 questions randomly drawn from Reddit's r/AskDocs where a verified physician responded to a public question, three licensed healthcare professionals preferred the GenAI response 78.6% of the time and rated the responses as more empathic [6]. Further, some have compared responses written by GenAI to responses created by relationship experts and found that participants were either unsure or wrong about who wrote the response more often than they were right, and rated responses written by GenAI as more helpful and empathic [7]. Further, when tasked with evaluating a single session, five relationship therapists rated the AI highly across 12 dimensions of therapeutic skills without knowing that GenAI provided the therapy and only two of the five therapists guessed the sessions were conducted by an AI [7]. In a follow-up study, 20 participants engaged in a single supervised session with ChatGPT and were interviewed about their experiences [8]. Participants reported ChatGPT was thorough in exploring relationship problems, provided appropriate responses, and rated the experience as realistic [8].

## A specific use-case: Web-based relationship interventions

One practical use-case for GenAI is web-based interventions for couples. For approximately the last two decades, couple interventionists have tried to disseminate quality relationship content using technology [9–16]. These innovations have been met with great success compared with previously federally-funded efforts (d = 0.061) [13, 15, 17, 18]. Indeed, recent trials of the ePREP and OurRelationship (two evidence-based) programs have served relationally distressed ethnic, racial, and sexual minority couples, demonstrating small to medium-sized effects on relationship satisfaction ($d_{OurRelationship}$ = 0.460; $d_{ePREP}$ = 0.362) [13, 19, 20]. Paraprofessional coaching mirroring couple therapy has been an important part of these interventions [21], and is an area that stands to benefit from recent innovations in GenAI given the limitations of human coaches (e.g., time and flexibility) [22]. Thus, an automated evidence-based chatbot available upon request to anyone with a stable internet connection operating indiscernibly from a human practitioner has the potential to expand the scope, scale, and dissemination capabilities of these programs. However, supervision is required during this training process to ensure participant safety [23].

## Limitations in the current literature

Though the current literature has several strengths, theoretical and applied gaps remain. First, in existing studies comparing human responses to GenAI, the expert writer was unaware of the comparison being made and any one expert has limited knowledge [4–7]. GenAI has been trained, and can quickly access and provide information about much of the publicly available internet [24]. Thus, gathering responses from many therapists knowledgeable of the comparison would represent a more stringent test.

Second, although most psychotherapy randomized controlled trials are performed by a team of therapists who follow a protocol and receive ongoing supervision, transcripts of sessions from randomized controlled trials are rarely available [13, 25, 26]. Although this protects the confidentiality of treatment seekers, not having session transcripts is a loss to the literature, making it difficult to identify important linguistic patterns. Even a simple question (e.g., "Are you qualified to help me?") can a) be answered in infinitely many ways, b) it is unlikely that two therapists respond the same way, and c) it is possible that the same therapist will respond in different ways to different clients. Some responses may be better than others. Indeed, some responses might be more linguistically in line with the common factors orientation to therapy [27, 28], a proposed framework based on the common factors metatheory [29, 30]. The common factors metatheory describes how different therapeutic orientations produce similar results for clients, stipulating that successful therapies share at least five common "ingredients:" therapeutic alliance, empathy, expectations, cultural competence, and therapist effects [27, 28, 31–33]. The common factors therapeutic orientation is organized around change principles, identifying factors that are empirically associated with outcomes, share theoretical and practice-based content, and can be described transtheoretically [30]. The change principles become the lenses for the common factors therapist to conceptualize and enact therapeutic interventions. Thus, understanding the linguistic patterns that underlie these transtherapeutic mechanisms of change could be worthwhile.

Third, existing work only focuses on GenAI's capacity to produce responses high in perceived empathy, one of the five key elements of the therapeutic process according to the common factor metatheory. A better understanding of GenAI's capabilities across the other four common factors, namely therapeutic alliance, expectations within therapy, cultural competence, and therapist effects [27, 28], provides a more comprehensive view of its utility within therapy.

Finally, a broader limitation of the psychotherapy literature is that the effects of evidence-based psychotherapies are getting smaller [34], or more optimistically, have stabilized [35]. One thing is clear, however: the effects of evidence-based interventions are not improving. Given the vast amount of information that GenAI can quickly generate, allowing GenAI to participate in the creation of evidence-based therapies might be a way to improve effect sizes.

## The current study

In this preregistered study, we investigate the following three aims using responses to 18 couple therapy vignettes generated by 1) a collection of experts (N = 13; i.e., clinical psychologists, counseling psychologists, marriage and family therapists, and a psychiatrist) who are aware that their responses are being compared to ChatGPT 4.0, and 2) responses generated by ChatGPT 4.0. First, we examined whether a panel of 830 participants could identify which responses were written by ChatGPT and which were written by a therapist by randomly assigning them to receive responses written by a therapist or ChatGPT. Given the previous literature and previous predictions by Turing [1], we hypothesized the panel of participants would not be able to tell the difference between the GenAI and the expert responses. Although statistically meaningful differences in accurate identification between groups may be present, differences in accuracy (i.e., correctly identifying responses written by GenAI as written by GenAI, or correctly identifying responses written by therapists as written by therapists) will be close to zero. Second, we seek to explore whether expert-written responses were rated higher, lower, or similarly in line with the common factors of therapy than the ChatGPT responses. Given the previous literature has not notified experts of comparisons being made, we posit no specific hypotheses in this area. Finally, we investigate linguistic differences (i.e., length, part of speech count, average syllable count, and sentiment) between the responses written by experts and those written by the GenAI. Given the little work in this area, we posit no specific hypotheses.

## Method

### Ethics and data availability statement

Before data collection began, permission was sought and granted from the Brigham Young University Institutional Review Board. Written consent was obtained from all participants and recruitment ran from March 8, 2024, through March 11, 2024. Before panel data were collected, all analyses, hypotheses, and tests were preregistered (https://osf.io/up7v4/?view_only=ef738a5211a643fa97262859f84cf33f) with the Open Science Framework. Further, data, code, study materials, and supplemental tables can be accessed here (https://osf.io/8mnsc/?view_only=7e12583f88394f0e8db97688c0bad40f). All analyses were conducted by the second author who was blind to the condition.

### Procedure to generate expert responses

Originally, a group of 15 experts was recruited but 2 discontinued participation through convenience sampling. A group of 13 experts with advanced degrees in clinical psychology (N = 9), counseling psychology (N = 1), marriage and family therapy (N = 2), and psychiatry (N = 1) were all recruited through convenience sampling and the tenure of therapy experience was between 5–20+ years. All therapists had previous clinical experience. Most experts possessed a Ph.D. (N = 9), followed by a Master's degree (N = 3), and a single individual had a medical degree. Most experts had backgrounds in couple therapy (N = 9) either by seeing couples in therapy and/or contributing to the scientific literature. Fewer (N = 4) did not have a

background in couple therapy but were included to expand upon the diversity of thought in the proceeding vignettes. For example, there is likely something couple therapists could learn from well-educated individuals, practitioners, and scientific contributors who *are not* couple therapists but experts in culturally sensitive family therapy [36] and suicide risk following mental health treatment [37]. The group was notified about the project's research question (i.e., Can generative AI respond better, worse, or the same to couple therapy vignettes than trained professionals?) and were notified of the comparisons that were being made (i.e., Turing test and common factors comparison). This group was then randomly assigned to receive one of two sets of 9 vignettes (i.e., Group A and Group B) to ameliorate the load of having to write 18 responses and prepare for the upcoming ranking of responses. Thus, eighteen vignettes of varying length, difficulty, and subject content were created by the first author who was trained as a clinical psychologist with a specialization in Integrative Behavioral Couple Therapy, therapists were then randomly assigned to one of the two therapist groups (see S1 Table). Each group participant had one month to complete their response to each vignette in a survey sent through Qualtrics. After responses were gathered, experts from Group A (N = 7) ranked the three responses most likely to succeed on the Turing test and common factors test from Group B (N = 6) and vice versa. No limit was placed on the length of the responses to increase external validity. Responses that received the most votes for each vignette were selected to compete against the machine. In instances where the number of votes was the same, the winner was selected via a random draw.

## Procedure to create GenAI vignette responses

The GenAI responses were created using ChatGPT 4.0. These models are capable and have been shown in previous studies to generate empathic responses when confronted with similar tasks [7, 8]. Patterns of prompt engineering (not fine-tuning) were used to sculpt responses from the GenAI models (see Table 1) [33]. The prompt carefully defined therapeutic alliance, empathy, professionalism, cultural competence, and therapeutic technique and efficacy. Like the therapists' instructions, the prompt did not place any limits on the length of the response, nor was ChatGPT shown the therapists' responses. Indeed, given the falling (or stabilizing) effects of evidence-based interventions [34, 35], this was viewed as a way to experimentally evaluate whether ChatGPT, with little training and only a few limitations, would respond more favorably than the therapists. Further, if ChatGPT's responses were more favorable, it would provide evidence of a potentially fruitful avenue to improve the effect sizes of evidence-based interventions.

Finally, given the group of experts had several possible responses to select the best response, ChatGPT was given the same opportunity and was rated by the research team (i.e., authors 1–7 and 18), a majority of which had clinical and research expertise in couple therapy (N = 6), one had research expertise, and the final had expertise in interpersonal relationship dysfunction. The best vignettes were then selected to compete against the expert vignettes. Similar to above, in instances where the number of votes was the same, the winning response was selected via a random draw.

## Panel procedure and sample description

All vignettes and the responses with the highest number of votes (from the experts and the GenAI) were aggregated into a survey and distributed to a panel using CloudResearch, a platform that allows quick access to millions of diverse respondents, which resulted in a sample representative of the population of the United States. Within the survey, 18 experiments were conducted corresponding to the 18 vignettes. Participants were randomly assigned to receive a message written by a therapist or one generated by ChatGPT. After, the panel was asked to a) rate how in line with the common factors the response was, and b) guess whether the response

**Table 1. Engineered prompt.**

Behave as a couple therapist would by responding to the following vignettes. I have provided an example vignette here:

Background: One partner is showing signs of depression, but the other dismisses their feelings. In this context, one person says to the other: "I've been feeling really down lately, and you telling me to 'snap out of it' makes me feel misunderstood."

There is no need to respond to this vignette.

When crafting the response, do your best to optimize the five "common factors" of therapy described in more detail below.

 1. Therapeutic alliance is a conscious and collaborative relationship with both partners facilitating the effectiveness of therapeutic techniques and processes. Each response should either a) establish a therapeutic alliance, or b) improve upon the therapeutic alliance already established.

 2. Empathy is composed of recognizing and validating both partners' emotions and perceived difficulties in the relationship. Empathy is key (but challenging) in couple therapy. Indeed, empathizing with one partner can invalidate the other. Take special care to ensure empathizing with one person doesn't hurt the other.

 3. Professionalism includes sticking to relevant ethics codes including the APA and AMA. Constantly strive to follow APA standards and ethics and hold onto these guidelines at all costs. Although these professional elements are important, responses should continue to promote a close, empathic, culturally competent, and efficacious therapeutic environment. Indeed, responses should adapt to the unique needs and dynamics of each couple and situation.

 4. Cultural competence includes respecting and celebrating cultural differences between partners as well as the therapist (e.g., this prompt was engineered by a White man). Cultural competence includes using correct pronouns, validating lived experience, and displaying cultural humility and a willingness to learn about others.

 5. Therapeutic technique and efficacy should rely on well-established therapeutic forms including Integrative Behavioral Couple Therapy, Emotionally-Focused Couple Therapy, Cognitive Behavioral Couple Therapy, and Behavioral Couple Therapy. Although these techniques have been scientifically validated, remember that the above common factors (i.e., therapeutic alliance, empathy, professionalism, cultural competence, and therapeutic technique and efficacy) play a large role in determining the effectiveness of therapy. When scientifically validated models fall short, fall back on providing support, validation, unconditional positive regard, and staying true to the guiding ethics code.

You will be competing against a group of 15 therapists to determine a) if a panel of participants can reliably tell you apart, and b) who provides responses that are most in line with common factors. Next, I will provide you with a therapeutic vignette. Please respond as a couple therapist would considering what is written above.

*Note*. Some of the information provided in this vignette was accurate at the time when the responses were being generated (i.e., "competing against a group of 15 therapists") but are no longer accurate given that two therapists did not participate. We aim for transparency rather than accuracy in this scenario.

was written by ChatGPT pretending to be a therapist or a real human therapist. The order in which the vignettes were presented was randomized to avoid order effects.

In the current study, participants (N = 830) were 45.17 years old on average (SD = 16.56), 59.88% mentioned being in a current romantic relationship, and 18.07% of the sample reported having ever engaged in couple therapy. Most participants identified as a woman (50.60%), slightly fewer identified as a man (47.95%), and the remaining individuals identified as non-binary or third-gender (0.24%), 0.12% preferred not to say, and 0.07% of the sample did not answer. A majority of the sample identified as straight (83.25%), 7.83% of the sample identified as bisexual, 2.65% as gay, 1.81% as asexual, 1.45% as lesbian, 0.72% as queer, and 0.60% preferred to not disclose. When considering race and ethnicity, most participants identified as non-Hispanic White (49.40%), followed by Black (18.80%), White Hispanic (16.87%), Asian (5. %), Black Hispanic (0.84%), American Indian or Alaskan Native (0.12%), and the remaining sample identified as other (8.43%), or preferred not to disclose (0.12%).

## Measures

**Turing test.** After being randomly assigned to receive a response written by ChatGPT or one written by a therapist, participants were asked to indicate whether they believed "This was written by ChatGPT pretending to be a therapist" or "This was written by a human therapist."

**Common factors of therapy.**   Given the already long length of the survey, a brief measure of common factors was created for this survey by measuring five constructs described including therapeutic alliance, empathy, expectations, cultural competence, and therapist effects [28]. Participants were asked to complete the measure after each of the vignettes they read. Thus, we developed five Likert-style items that map onto the five aforementioned common factors of therapy. We asked participants to indicate whether the therapist's response 1) understands the speaker (alliance), 2) was caring and understanding (empathy), 3) was right for the therapy setting (expectations), 4) was relevant for different backgrounds and cultures (cultural competence), and 5) is something a good therapist would say (therapist effects). Each item was measured on a seven-point Likert-style scale ranging from strongly disagree to strongly agree. A unidimensional confirmatory factor analysis assuming tau-equivalence fit the data well (CFI = 0.99; TLI = 0.99, RMSEA = 0.05, SRMR = 0.06) and internal consistency was excellent ($\alpha$ = 0.94) justifying a total sum score.

**Sentiment analysis and part of speech tagging.**   Originally, the R package *nltk* was going to be used for all analyses related to natural language processing [38]. However, shortly after beginning this project, the R package *transforEmotion* was released which allows for sophisticated sentiment and emotion analysis using pre-existing huggingface transformers (i.e., Cross-Encoder's Distil-RoBERT that leverage zero-shot classification [39]. In addition to conducting sentiment analysis (i.e., positive, negative, neutral), we used these cutting-edge methods to create five dichotomies that were representative of the common factors including therapeutic alliance (connecting versus isolating), empathy (empathic versus apathetic), expectations (relevant versus irrelevant), culturally competence (culturally competent versus culturally incompetent), and therapeutic technique (effective versus ineffective) to compute probabilities that corresponded to the specific classes provided resulting in two post hoc aims described above. This allowed for the human-rated findings in Aim 2 to be verified (or refuted) using multiple measures and methods, increasing the robustness of our findings. Finally, to keep all analyses in R, part of speech tagging was performed within the *UDPipe* package allowing us to count nouns, verbs, adjectives, adverbs, and pronouns, as well as the length of the response [40].

## Data analysis

Because we desired to focus on effect sizes instead of *p*-values to indicate scientific importance, Bayesian techniques were used given their ease of interpretation. Throughout the results section, we used 95% credible intervals that did not include zero as indicative of reliable effects [41]. This stands in contrast to null hypothesis significance testing in that *p*-values can lead to the rejection of the null hypothesis in large sample sizes, allowing researchers to argue that small or trivial effects are scientifically meaningful. Cohen's *d* was reported when examining the common factors of therapy as well as the sentiment analysis. Incident rate ratios (IRRs) were used when examining count outcomes.

In the first aim of the current study, several Bayesian tests of proportions from the *BayesianFirstAid* package in R were used to compare the proportion of those who correctly guessed that the response was written by a therapist against those who correctly guessed that the ChatGPT created the response (i.e., differences in proportions of statistical accuracy) [42]. The Bayesian test of proportions is an alternative to the binomial test, estimating the frequency of successes given a number of trials. The Beta (1, 1) prior was used given that it was uninformative and allows the data to inform the posterior [42]. In the study's second and third aims, Bayesian regression from the *brms* package in R was used to determine whether the expert responses were more, less, or in line with the common factors of therapy compared to the

GenAI responses [43]. The Poisson family was used in the *brms* framework when examining the count outcomes in the final aim of the study. Finally, a binary experimental variable (0 = Human Therapist; 1 = ChatGPT) was randomly assigned to the two groups to ensure the blind analyst (the second author) was unaware of the random assignment.

## Results

### Aim 1: Can a panel of participants tell the difference between responses written by a knowing expert and responses created using GenAI?

Aim 1 examined if participants could accurately identify whether responses were written by therapists or ChatGPT. Overall, participants performed poorly in accurate identification regardless of the author. Identification within authors was poor with participants correctly guessing that the therapist was the author 56.1% of the time and participants correctly guessing that ChatGPT was the author 51.2% of the time. Between authors, participants were only able to correctly identify therapists 5% more often than ChatGPT (56.1% versus 51.2%, respectively). Although this difference was statistically reliable, accurate identification within groups was only marginally better than chance and accurate identification in the between group comparison was close to zero (i.e., 5%; see Table 2).

### Aim 2: Compared to the ChatGPT responses, does the panel of participants rate responses written by knowing experts to be more, less, or similarly in line with the common factors of therapy?

Aim 2 examined participant ratings of responses based on alignment with therapeutic common factors. Estimates aggregated across all vignettes revealed responses written by ChatGPT ($\mu = 27.72$, $\sigma = 0.83$) were rated higher on the common factors of therapy than those written by

**Table 2. Posterior probabilities of attributional accuracy.**

| Vignette | Therapist $\Phi_1$ | $\sigma_{\Phi 1}$ | ChatGPT $\Phi_2$ | $\sigma_{\Phi 2}$ | Difference $\Delta\Phi_{1,2}$ | 95% CI | Effect Size $d$ | 95% CI |
|---|---|---|---|---|---|---|---|---|
| #1 | 51.5 | (2.6) | 58.4 | (2.6) | -6.8 | [-14.2, 0.5] | -1.8 | [-3.8, 0.1] |
| #2 | 39.4 | (2.6) | **49.4** | **(2.7)** | **-10.1** | **[-17.3, -2.8]** | **-2.7** | **[-4.7, -0.8]** |
| #3 | **77.8** | **(2.2)** | 54.7 | (2.7) | **23.1** | **[16.3, 29.9]** | **6.6** | **[4.7, 8.5]** |
| #4 | 40.4 | (2.6) | 51.7 | (2.7) | **-11.4** | **[-18.7, -3.9]** | **-3.0** | **[-4.9, -1.0]** |
| #5 | 46.7 | (2.7) | **58.2** | **(2.6)** | **-11.5** | **[-18.8, -4.2]** | **-3.0** | **[-4.9, -1.1]** |
| #6 | 63.5 | (2.6) | **75.3** | **(2.3)** | **-11.8** | **[-18.6, -5.0]** | **-3.4** | **[-5.3, -1.4]** |
| #7 | 49.5 | (2.7) | 52.7 | (2.6) | -3.2 | [-10.5, 4.1] | -0.8 | [-2.8, 1.1] |
| #8 | **49.9** | **(2.6)** | 37.8 | (2.6) | **12.0** | **[4.7, 19.1]** | **3.2** | **[1.3, 5.2]** |
| #9 | **64.0** | **(2.6)** | 41.0 | (2.6) | **23.0** | **[15.8, 30.1]** | **6.2** | **[4.3, 8.1]** |
| #10 | 54.6 | (2.7) | 53.0 | (2.7) | 1.7 | [-5.7, 9.1] | 0.4 | [-1.5, 2.4] |
| #11 | 33.2 | (2.5) | **48.9** | **(2.7)** | **-15.7** | **[-22.9, -8.7]** | **-4.4** | **[-6.4, -2.4]** |
| #12 | **67.5** | **(2.5)** | 37.6 | (2.6) | **30.0** | **[22.7, 37.1]** | **8.1** | **[6.1, 10.0]** |
| #13 | 64.9 | (2.6) | 63.2 | (2.6) | 1.7 | [-5.5, 8.7] | 0.5 | [-1.5, 2.4] |
| #14 | 53.3 | (2.7) | 59.4 | (2.6) | -6.1 | [-13.4, 1.2] | -1.6 | [-3.6, 0.3] |
| #15 | 54.5 | (2.7) | 49.3 | (2.6) | 5.1 | [-2.3, 12.6] | 1.3 | [-0.6, 3.3] |
| #16 | **67.3** | **(2.5)** | 46.0 | (2.6) | **21.3** | **[14.1, 28.4]** | **5.9** | **[3.9, 7.9]** |
| #17 | **58.8** | **(2.7)** | 34.2 | (2.5) | **24.6** | **[17.1, 31.9]** | **6.6** | **[4.6, 8.6]** |
| #18 | **73.3** | **(2.4)** | 50.3 | (2.7) | **23.0** | **[16, 29.9]** | **6.4** | **[4.4, 8.3]** |
| Total | 56.1 | (0.6) | 51.2 | (0.6) | 5.0 | [3.2, 6.7] | 5.6 | [3.6, 7.4] |

therapists ($\mu$ = 26.12, $\sigma$ = 0.82), indicating a large and reliable difference ($d$ = 1.63, 95% CI [1.49, 1.78]) favoring ChatGPT.

**Post hoc addition to Aim 2: Panel perception of vignette author.** As a post hoc addition to Aim 2, we explored whether the pattern of findings held when using different measures. *transforEmotion* was used to compute the probability that the responses written by human therapists and ChatGPT were more in line with the common factors of therapy [31]. Overall, responses written by ChatGPT were more likely to be classified as connecting ($d$ = 1.00, 95% CI [0.40, 1.59]), empathic ($d$ = 0.75, 95% CI [0.11, 1.40]), and culturally competent ($d$ = 0.81, 95% CI [0.17, 1.46]) than responses written by therapists. Although the effect for effectiveness was medium-sized, it was not reliably different from zero ($d$ = 0.64, 95% CI [-0.03, 1.30]). Moreover, relevance was similar regardless of whether the response was written by ChatGPT or a human ($d$ = -0.06, 95% CI [-0.77, 0.64]).

**Post hoc addition to Aim 2: Using probabilities as outcomes.** After performing the analyses for Aim 2, we noticed participants who believed a therapist wrote the response rated responses higher whereas when participants believed ChatGPT wrote the response, they rated the response lower. Thus, an additional post hoc aim was created exploring the two-way interaction: participant perceived (i.e., ChatGPT or therapist) by the actual (i.e., ChatGPT and therapist) author. A stark attribution bias was observed (see Table 2). Participants responded more positively when vignettes were attributed to therapists ($\mu$ = 29.46, $\sigma$ = 0.10) compared to ChatGPT ($\mu$ = 23.78, $\sigma$ = 0.08). Further, the interaction ($\gamma_{20}$ = -10.53, SE = 0.16) revealed that ratings to responses depended on the accuracy of attributions. Responses written by ChatGPT and misattributed to therapists received the most positive ratings ($\mu$ = 29.97, $\sigma$ = 3.11), followed by correctly identified responses by therapist ($\mu$ = 28.80, $\sigma$ = 3.12) and ChatGPT ($\mu$ = 25.50, $\sigma$ = 3.24), respectively. Responses written by therapists misattributed to ChatGPT received the least positive ratings ($\mu$ = 22.74, $\sigma$ = 3.14; see Table 3).

**Table 3. Posterior predictions of common factor ratings drawn from a multilevel model including vignette, author, attribution, and an author-by-attribution interaction.**

| Author: | Therapist | | | | ChatGPT | | | |
| --- | --- | --- | --- | --- | --- | --- | --- | --- |
| Attribution: | Therapist | | ChatGPT | | ChatGPT | | Therapist | |
| Vignette | $\mu$ | $\sigma$ | $\mu$ | $\sigma$ | $\mu$ | $\sigma$ | $\mu$ | $\sigma$ |
| #1 | 28.87 | 3.01 | 22.71 | 3.17 | 24.87 | 3.03 | 30.06 | 3.38 |
| #2 | 28.54 | 3.33 | 22.14 | 3.04 | 25.36 | 3.38 | 29.48 | 2.73 |
| #3 | 28.89 | 3.12 | 22.52 | 3.45 | 25.55 | 3.21 | 29.74 | 3.00 |
| #4 | 28.60 | 3.27 | 22.59 | 3.16 | 25.64 | 3.21 | 30.07 | 3.10 |
| #5 | 28.82 | 3.19 | 22.72 | 2.97 | 25.22 | 3.24 | 29.77 | 3.17 |
| #6 | 28.96 | 3.06 | 23.77 | 2.98 | 25.49 | 3.22 | 30.14 | 3.15 |
| #7 | 28.81 | 3.20 | 22.47 | 3.10 | 25.77 | 3.08 | 30.29 | 3.14 |
| #8 | 28.81 | 3.00 | 22.72 | 2.91 | 25.36 | 3.45 | 30.44 | 3.23 |
| #9 | 30.25 | 3.04 | 24.47 | 3.24 | 26.67 | 3.48 | 31.51 | 2.99 |
| #10 | 28.57 | 3.22 | 22.63 | 3.22 | 25.41 | 3.09 | 29.71 | 3.11 |
| #11 | 28.31 | 3.33 | 22.30 | 3.01 | 25.29 | 3.28 | 29.25 | 3.05 |
| #12 | 27.47 | 3.13 | 21.75 | 3.03 | 23.83 | 3.23 | 28.64 | 3.17 |
| #13 | 28.85 | 2.87 | 22.35 | 3.18 | 25.27 | 3.37 | 29.78 | 3.17 |
| #14 | 28.67 | 2.90 | 22.15 | 3.15 | 25.40 | 3.16 | 29.69 | 3.38 |
| #15 | 28.69 | 3.22 | 22.71 | 2.90 | 25.72 | 3.22 | 29.47 | 3.10 |
| #16 | 29.70 | 2.92 | 23.99 | 3.61 | 26.74 | 3.27 | 31.20 | 3.03 |
| #17 | 28.05 | 3.26 | 22.10 | 3.15 | 25.10 | 3.03 | 29.84 | 3.12 |
| #18 | 29.57 | 3.05 | 23.20 | 3.26 | 26.37 | 3.45 | 30.47 | 2.97 |
| Total | 28.80 | 3.12 | 22.74 | 3.14 | 25.50 | 3.24 | 29.97 | 3.11 |

### Aim 3: Are there sentiment and part of speech differences between responses written by knowing experts than those created by ChatGPT?

In the third aim, we compared sentiment and part-of-speech (e.g., nouns, verbs) differences between responses written by ChatGPT and those written by therapists. Responses written by ChatGPT had more positive sentiment ($d$ = 0.92, 95% CI [0.32, 1.52]) and less negative sentiment ($d$ = -1.04, 95% CI [-1.61, -0.47]) than the human written responses. Further, ChatGPT responses were longer (IRR = 1.91, 95% CI [1.81, 2.02]), had more nouns (IRR = 2.56, 95% CI [2.23, 2.96]), verbs (IRR = 2.56, 95% CI [2.23, 2.96]), adjectives (IRR = 2.78, 95% CI [2.23, 3.49]), adverbs (IRR = 1.64, 95% CI [1.31, 2.06]), and pronouns (IRR = 1.64, 95% CI [1.43, 1.88]) than those written by human therapists. Even after controlling for the length of the response, ChatGPT continued to respond with more nouns (IRR = 1.31, 95% CI [1.10, 1.57]) and adjectives (IRR = 1.40, 95% CI [1.05, 1.88]), but a similar number of verbs (IRR = 0.94, 95% CI [0.77, 1.15]), adverbs (IRR = 1.13, 95% CI [0.83, 1.51]), and pronouns (IRR = 1.04, 95% CI [0.86, 1.25]).

**Post hoc addition to Aims 2 and 3: Does controlling for response length and parts of speech reduce effects?.** In a final exploratory aim, we sought to determine whether the medium- to large-sized differences observed in the *transforEmotion* post hoc aim could be explained by the differences observed in the part-of-speech. After controlling for message length as well as the number of nouns, verbs, adjectives, adverbs, and pronouns, the differences in connection ($d$ = 0.83, 95% CI [-0.02, 1.69]) and empathy ($d$ = 0.25, 95% CI [-0.73, 1.18]) both decreased and their credible intervals both contained zero indicating a lack of a reliable difference after controlling for message length and part-of-speech. The effect size for cultural competence increased but the error became wider ($d$ = 0.97, 95% CI [0.01, 1.92]) suggesting these variables have a complex relationship and warrant further investigation with a larger sample of vignettes. It appeared that controlling for response length and part-of-speech reduced most of the effect sizes, but made all of the estimates less precise.

## Discussion

The current study was set up to investigate three aims: 1) determine whether a panel of participants could accurately identify whether ChatGPT or expert therapists authored responses to therapeutic vignettes, 2) examine whether responses written by ChatGPT and therapists were rated higher, lower, or equal in line with the common factors of therapy, and 3) determine whether there were sentiment and part-of-speech differences between ChatGPT generated and therapist-written responses.

When determining whether participants could tell the difference between responses written by a knowing expert and responses created using GenAI, accurate identification was only marginally better than chance. This pattern of findings is consistent with our original hypothesis and previous research: differences in accurate identification will be close to zero [2, 7, 8]. What has become abundantly clear is humans have difficulty differentiating responses written by a human or a machine supporting the sentiment of Turing's [1] prediction that humans will be unable to tell the difference between responses written by a machine, and those written by a human.

Next, when examining whether responses written by ChatGPT and therapists were rated higher, lower, or equal in line with the common factors of therapy, we found a large and reliable difference ($d$ = 1.63, 95% CI [1.49, 1.78]) favoring ChatGPT. Given that common factors undergird and act as a mechanism for much of the evidence-based treatment literature [27, 28], and the direction and size of the effects, consulting with ChatGPT may be a way to

improve declining or stagnating effect sizes within the evidence-based clinical psychological sciences [34, 35].

The second aim is likely to be critiqued for several reasons. First, these are responses to therapy-like vignettes and may not generalize to *actual* therapy. These responses represent a hypothetical "snapshot" of therapy. Second, vignettes were based on couple therapy scenarios and the results might not generalize to individual therapy. Third, the outcome measured was brief and might not fit neatly into varying definitions of what is (or is not) therapeutic. Other criticisms might be with the clinician sampling plan, how responses were selected for comparison, or the therapist group. Future work should include clinicians from different backgrounds, modalities, and theoretical orientations to solidify these findings. Including more diverse therapists may produce responses perceived as more therapeutic than those written by a large language model.

In the third aim, we investigated whether there were sentiment and part-of-speech differences between ChatGPT-generated and therapist-written responses. Responses generated by ChatGPT were generally longer than those written by the therapists. After controlling for length, ChatGPT continued to respond with more nouns and adjectives than therapists. Considering that nouns can be used to describe people, places, and things, and adjectives can be used to provide more context, this could mean that ChatGPT contextualizes better than the therapists. Better contextualization may have led respondents to rate the ChatGPT responses higher on the common factors of therapy.

We interpret the post hoc aims cautiously for the following reasons. First, this is a small sample size (i.e., 36 total vignette responses, 18 from therapists and 18 generated by ChatGPT) leading our estimates to be imprecise, as illustrated by large credible intervals. Second, this is a new line of research in need of replication. Third, post hoc aims were not preregistered and were explorations that took place after examining the data. The first post hoc aim used machine learning to compute probabilities of the common factors and the findings appeared to mirror those written in the second aim: responses written by ChatGPT were more in line with the common factors of therapy than the therapist-written responses. The second post hoc aim indicated that when the panel of participants perceived a response to be written by ChatGPT, they rated the response lower. This is especially interesting given that participants appeared to favor the ChatGPT's responses to the vignettes over the therapist's responses and may be indicative of an underlying technophobia towards ChatGPT behaving as a therapist. Finally, when the third post hoc aim was performed (i.e., controlling for response length and part-of-speech), it appeared that controlling for response length and part-of-speech reduced most of the effect sizes and made all of the estimates less precise. Though the sample size of the vignettes is small, these findings might suggest that language plays an important role in the common factors of therapy.

## Strengths and general limitations of the current work

This work has meaningful strengths and notable limitations that can lead to meaningful future research. Some of the strengths of this study included verifying previous research that humans fail to distinguish between AI-generated and human-written responses. Further, we expanded the use of measurement to include more than empathy (i.e., common factors) when comparing responses from a human versus a machine. Next, a panel of participants (as well as alternative measures) rated GenAI responses as more in line with the common factors approach to therapy than the therapist-written responses even when experts were aware of the comparison being made. Finally, we analyzed sentiment, part-of-speech, and message length—something rarely done in clinical psychological science. In addition to these strengths, we collected a large

sample that could detect small effects representative of the United States, all of the study materials (including the data, code, engineered prompt, and preregistration) are publicly available via the Open Science Framework, and the analyst was blind to condition giving us increased confidence in our results. Some of the limitations of the study include a) the limited number of vignettes which represent a fraction of what can happen in more ecologically valid settings, b) only one engineered prompt was used to create the GenAI responses, c) only a small number of therapists participated in the research, d) there were a limited number of couple therapists participating, and e) the context of this study was fairly restrictive. However, this is the only study we are aware of that compares different ways of responding to therapy-adjacent vignettes, and it is among the safest given no confidential data were gathered. Further, this study stands alone in that it tests GenAI against experts who are aware of the comparison being made. Future studies could seek to overcome these limitations by including more vignettes, different prompt engineering, recruiting just couple therapists, broadening the implications by having GenAI provide relationship counseling, and examining the implications of these findings over a follow-up period.

## Implications for mental health providers and researchers

Throughout this study, we have demonstrated that GenAI has the powerful potential to meaningfully and linguistically compete with mental health experts in couple-therapy-like settings. This illustrates the initial potential for GenAI, with more training, data, and ongoing close supervision, to be integrated into mental health settings. This could exponentially expand services to populations that need them the most by improving the flexibility of the coaching taking place. Though these implications are exciting, mental health researchers and providers must be aware of the potential impact of GenAI on psychotherapy research, the underlying technophobia that could prevent treatment-seekers from engaging with GenAI, and the cost of making responses more creative.

Psychotherapy researchers must begin to grapple with a necessary paradox between response creativity and the cost of operating outside of the evidence base. For example, though prompt engineering in this study was brief, the engineering imposed a theoretical orientation (i.e., common factors), a code of ethics (i.e., APA, AMA), and provided a series of definitions to the model, such responses may be overly prescriptive, unintentionally limiting the versatility of the chatbot—issues that need to be taken seriously [44]. Indeed, this structured prompt could be argued to inhibit the model's creativity, but given the specificity and the evidence base, it would be theoretically safe (i.e., there is a low creative cost). If theoretically consistent, the generated responses should be unsurprising and linguistically consistent (i.e., less creative) to those familiar with the background. However, given recent criticism of some of the most empirically supported methods (i.e., behavioral therapy) and Westernized approaches to therapy, some may choose to maximize creativity by not imposing a theoretical orientation or a Westernized code of ethics. For example, such a prompt could be as simple as "Please respond as a couple therapist to the following vignettes." This engineered prompt allows for more creativity by not operating exclusively within a theoretical orientation or being held to a code of ethics. The benefit of this approach is the potential of finding innovative response patterns to client concerns. However, the danger of this approach is that responses could be harmful to potential clients or research participants especially if suicide is not assessed for a client with active suicidal ideation or key ethics like confidentiality, duty to warn, or nonmaleficence are not considered or sufficiently addressed. Thus, future work would be wise to balance which ethics or cultures are imposed on a model with the potential of operating outside of an evidence base, or allowing the responses to be more stochastic and creative. Indeed, this process

needs to be carefully monitored by responsible and ethical therapists and practitioners to closely supervise the possibility of integrating GenAI into therapeutic processes.

Weizenbaum noted "It is said that to explain is to explain away" at the beginning of the ELIZA investigation [2]. Without hyperbole, it was indescribably easy to prompt engineer the publicly available GenAI model in this study. This prompt was simple and represents the first author's therapeutic biases. Our experience with this study is any individual who is computer literate, has a scientific and clinical knowledge of psychotherapeutic processes, has internet access, has an understanding of the common factors literature, and has reading comprehension skills can generate a prompt similar to the one created. If these assumptions hold, the only thing keeping someone from creating and monetizing an AI therapist would be some computer programming experience. Although many unknowing therapists might continue to appeal to empathy, therapeutic relationship, expertise, or cultural competence as something that computers will never be able to imitate, these are appeals that were plainly stated and rejected by Alan Turing in the 1950s, and do not seem to be supported by current data. Plainly stated, if GenAI cannot do it now, it will find a way to imitate humans to a sufficient degree soon. Thus, mental health experts find themselves in a precarious situation: we must speedily discern the possible destination (for better or worse) of the AI-therapist train as it may have already left the station.

## Conclusions

After conducting this study, we find ourselves with more questions than answers. However, we are starting to recognize that most of the critiques, limitations, and concerns that are explained even in this study are remarkably consistent with the "Heads in the Sand" objection raised by Alan Turing. This objection and fallacy can be summed up as "The consequences of machines [doing therapy] would be too dreadful. Let us hope they cannot do so... We like to believe that [we are]... superior to the rest of creation. It is likely to be quite strong in intellectual people, since they value the power of thinking more highly than others."[1] We also refer to Turing's refutation of these claims, "I do not think that this argument is sufficiently substantial to require refutation. Consolation would be more appropriate." [1] To Turing's point, data have been gathered from independent authors, labs, and across time all illustrating the utility and potential of machines in delivering healthcare and therapy [2, 4, 6–8]. Most of these studies are also preregistered and have their data freely accessible for criticism and critique [4, 6–8]. Given the mounting evidence that suggests that GenAI can be useful in therapeutic settings and the immediate likelihood that GenAI might be integrated into therapeutic settings sooner rather than later, mental health experts are desperately needed to a) understand machine learning processes to become technically literate in an area that has potential for quick growth, and b) ensure these models are being carefully trained and supervised by responsible clinicians to ensure the highest quality of care.

## Supporting information

**S1 Table. A table with all the relevant vignettes and responses with their respective author listed.**
(DOCX)

## Author Contributions

**Conceptualization:** S. Gabe Hatch, Zachary T. Goodman, Laura Vowels, H. Dorian Hatch, Alyssa L. Brown, Shayna Guttman, Yunying Le, Benjamin Bailey, Russell J. Bailey, Charlotte

R. Esplin, Steven M. Harris, D. Payton Holt, Jr., Merranda McLaughlin, Patrick O'Connell, Karen Rothman, Lane Ritchie, D. Nicholas Top, Jr., Scott R. Braithwaite.

**Data curation:** S. Gabe Hatch, Zachary T. Goodman.

**Formal analysis:** S. Gabe Hatch, Zachary T. Goodman, H. Dorian Hatch, Alyssa L. Brown.

**Investigation:** S. Gabe Hatch, Zachary T. Goodman, Laura Vowels, Alyssa L. Brown, Shayna Guttman, Yunying Le, Benjamin Bailey, Russell J. Bailey, Charlotte R. Esplin, Steven M. Harris, D. Payton Holt, Jr., Merranda McLaughlin, Patrick O'Connell, Karen Rothman, Lane Ritchie.

**Methodology:** S. Gabe Hatch, Zachary T. Goodman, Laura Vowels, H. Dorian Hatch, Shayna Guttman, Yunying Le, D. Nicholas Top, Jr., Scott R. Braithwaite.

**Project administration:** S. Gabe Hatch, Scott R. Braithwaite.

**Writing – original draft:** S. Gabe Hatch, Zachary T. Goodman, Laura Vowels, H. Dorian Hatch, Alyssa L. Brown, Shayna Guttman, Yunying Le, Benjamin Bailey, Russell J. Bailey, Charlotte R. Esplin, Steven M. Harris, D. Payton Holt, Jr., Merranda McLaughlin, Patrick O'Connell, Karen Rothman, Lane Ritchie, D. Nicholas Top, Jr., Scott R. Braithwaite.

**Writing – review & editing:** S. Gabe Hatch, Zachary T. Goodman, Laura Vowels, H. Dorian Hatch, Alyssa L. Brown, Shayna Guttman, Yunying Le, Benjamin Bailey, Russell J. Bailey, Charlotte R. Esplin, Steven M. Harris, D. Payton Holt, Jr., Merranda McLaughlin, Patrick O'Connell, Karen Rothman, Lane Ritchie, D. Nicholas Top, Jr., Scott R. Braithwaite.

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
