## [Decision Letter · Decision Letter 0]

30 Jul 2024

PMEN-D-24-00239

When ELIZA Meets Therapists: A Turing Test for the Heart and Mind

PLOS Mental Health

Dear Dr. Hatch,

Thank you for submitting your manuscript to PLOS Mental Health. After careful consideration, we feel that it has merit but does not fully meet PLOS Mental Health’s publication criteria as it currently stands. Therefore, we invite you to submit a revised version of the manuscript that addresses the points raised during the review process.

We look forward to receiving your revised manuscript.

Kind regards,

Brian Bauer, PhD

Academic Editor

PLOS Mental Health

Journal Requirements:

Additional Editor Comments (if provided):

Please find enclosed three reviews of your manuscript entitled, "When ELIZA Meets Therapists: A Turing Test for the Heart and Mind." Each of the reviewers is a highly experienced researcher in areas relevant to the topic that you address. Nonetheless, the reviewers are consistent in noting a number of ways in which your manuscript could be clarified or modified. Thus, after having read the reviewers’ comments as well as the manuscript in detail, my decision is for you to submit a revised manuscript. In your revision, it will be essential that you respond to each of the issues raised by the reviewers. At that time, your manuscript will be sent out for an additional round of reviews.

Reviewers' comments:

Reviewer's Responses to Questions

**Comments to the Author**

1. Does this manuscript meet PLOS Mental Health’s publication criteria? Is the manuscript technically sound, and do the data support the conclusions? The manuscript must describe methodologically and ethically rigorous research with conclusions that are appropriately drawn based on the data presented.

Reviewer #1: Yes

Reviewer #2: No

Reviewer #3: Yes

2. Has the statistical analysis been performed appropriately and rigorously?

Reviewer #1: Yes

Reviewer #2: I don't know

Reviewer #3: Yes

3. Have the authors made all data underlying the findings in their manuscript fully available (please refer to the Data Availability Statement at the start of the manuscript PDF file)?

Reviewer #1: Yes

Reviewer #2: Yes

Reviewer #3: Yes

4. Is the manuscript presented in an intelligible fashion and written in standard English?

Reviewer #1: Yes

Reviewer #2: Yes

Reviewer #3: Yes

5. Review Comments to the Author

Reviewer #1: The authors have done an excellent job of investigating a unique question with innovative methods. The quality of the writing, research design, and statistical analysis is to a high standard, and the rationale behind these components (e.g., the unique choice of choosing Bayesian techniques over frequentist ones) is clear.

Comment: While I was caught off guard by having the materials and method sections at the end of the paper, I could see afterward how that was an appropriate decision given the length and detail required for it. I assume the journal allows for this, and if so think it should be appropriate to keep - as interested readers will scroll down to do a close read of it. The authors did a terrific job of explaining the methods and materials used in the study and making it reproducible by having all the details available on OSF.

Suggestion: The only suggestion I would make is to introduce the concept of and literature behind, the common factors in more detail in the limitations in the literature section, given it is a key component of the analysis. It probably doesn't need its own section, but the paper might benefit from making the introduction of this concept more clear so that readers are not going back and doing a double take once its importance is made clear. I am hesitant to suggest that the authors add it before that limitations section, given it will ruin the flow, but if they have an idea of how to emphasize it and key papers introducing it, that would strengthen the paper.

Overall, I would like to reiterate that the paper is of a high standard, and I look forward to seeing it in print. I hope it will contribute to more research in this area as new techniques and tools are developed. I hope it also gets some media attention.

Reviewer #2: The article is strongly motivated with clear objectives. Evaluations comparing AI and human generations are increasingly important across clinical tasks, and the authors contribute to a growing body of research exploring this by focusing on the therapeutic setting. However, the article could be made more compelling by improved organization, more methodological detail, and more precise language. Further, the analysis corresponding to Aim 1 of the article may be flawed.

Weaknesses:

- Structure is confusing and a bit difficult to follow. Specifically, it is strange to see the methods section follow the conclusion. The methods section contains many important details that are necessary to engage with the results and discussion meaningfully. Additionally, including the results tables in the results text would make the article more readable.

- Method is unclear or insufficiently thorough. I think a reader would struggle to reproduce your work. The statistical modeling decisions could use further detail and justification. For example, the authors could better justify why Bayesian methods were particularly fitting for this problem. There is a sentence describing the general benefits of Bayesian modeling, but the method would be more compelling were the authors to explain why Bayesian methods were particularly fitting for this study. The authors also mention that several tests of proportions are used for the study. What are they?

- Small number of generations to compare, though the authors acknowledge this.

Strengths:

- The motivation is clear and easy to follow. The juxtaposition of this work in the Turing test context is interesting. The 3 aims help streamline the article.

- Experimental design is simple and reasonable.

- Strong significant results about the effectiveness of GenAI for therapy generation that should be relevant for the community of therapists, social workers, doctors, and researchers interested in understanding what role generative models should play in mental health services.

Other comments:

- Paper should be proofread. Several grammatical errors and incomplete sentences.

- The “5% more often” statistic for the results from Aim 1 is misleading. This statistics describes the difference in accuracy for ground truth human generations and ground truth AI generations. The language of the sentence containing this statistic makes it sound like the bad performance is attributed to there being a difference in these accuracy metrics, which is not the case. For example, recording 100% on both the set of human generations and set of AI generations would be ideal performance, but result in a difference of 0%. The relevant statistic here is the overall accuracy. If the authors want to discuss how that accuracy is broken down across different ground truth labels, then they should present the full confusion matrices.

- Following the above, I am not sure if the analysis done in Table 1 is correct. The analysis focuses on differences in probabilities across the difference ground truth labels, which may be fundamentally flawed.

- The phrase "statistically reliable" is used when presumably the authors mean statistically significant.

- The use of the phrase "trend toward" to describe point estimates is misleading. I assume the authors intend to say that the differences are "close to" 0 but do not have a way to communicate that technically/statistically. Again, this could be an issue interpreting the differences (which communicate little information) when instead the authors should be analyzing overall accuracy.

Reviewer #3: Thank you for this paper. I found this article very intriguing and exciting to read. In terms of cultural competency-I am a bit unclear on how this was determined by AI/therapists. Did the therapist/AI discuss cultural backgrounds/issues? In terms of widening creativity and not being limited to theory and ethics, how do you suggest the field use AI to ensure important factors such as confidentiality, duty to warn, nonmaleficence, etc is addressed.? Although it seems limitless using AI, does that really equate to creativity if for example, risk of suicide is not addressed by a machine?

6. PLOS authors have the option to publish the peer review history of their article (what does this mean?). If published, this will include your full peer review and any attached files.

**Do you want your identity to be public for this peer review?** For information about this choice, including consent withdrawal, please see our Privacy Policy.

Reviewer #1: No

Reviewer #2: No

Reviewer #3: No

---

## [Decision Letter · Decision Letter 1]

29 Oct 2024

PMEN-D-24-00239R1

When ELIZA Meets Therapists: A Turing Test for the Heart and Mind

PLOS Mental Health

Dear Dr. Hatch,

My apologies, this journal requires authors to address minor edits before an acceptance can be provided. Please disregard my previous message and make the edits/corrections to the address the second round of reviewers and resubmit when ready. Please let me know if you have any questions, and my apologies again for the confusion.

Thank you for submitting your manuscript to PLOS Mental Health. After careful consideration, we feel that it has merit but does not fully meet PLOS Mental Health’s publication criteria as it currently stands. Therefore, we invite you to submit a revised version of the manuscript that addresses the points raised during the review process.

Please submit your revised manuscript by . If you will need more time than this to complete your revisions, please reply to this message or contact the journal office at mentalhealth@plos.org. Please include the following items when submitting your revised manuscript:

We look forward to receiving your revised manuscript.

Kind regards,

Brian Bauer, PhD

Academic Editor

PLOS Mental Health

Journal Requirements:

Additional Editor Comments (if provided):

Reviewers' comments:

Reviewer's Responses to Questions

**Comments to the Author**

1. If the authors have adequately addressed your comments raised in a previous round of review and you feel that this manuscript is now acceptable for publication, you may indicate that here to bypass the “Comments to the Author” section, enter your conflict of interest statement in the “Confidential to Editor” section, and submit your "Accept" recommendation.

Reviewer #3: (No Response)

Reviewer #4: (No Response)

Reviewer #5: All comments have been addressed

Reviewer #6: (No Response)

2. Does this manuscript meet PLOS Mental Health’s publication criteria? Is the manuscript technically sound, and do the data support the conclusions? The manuscript must describe methodologically and ethically rigorous research with conclusions that are appropriately drawn based on the data presented.

Reviewer #3: Yes

Reviewer #4: (No Response)

Reviewer #5: Yes

Reviewer #6: Yes

3. Has the statistical analysis been performed appropriately and rigorously?

Reviewer #3: Yes

Reviewer #4: (No Response)

Reviewer #5: I don't know

Reviewer #6: Yes

4. Have the authors made all data underlying the findings in their manuscript fully available (please refer to the Data Availability Statement at the start of the manuscript PDF file)?

Reviewer #3: Yes

Reviewer #4: (No Response)

Reviewer #5: Yes

Reviewer #6: Yes

5. Is the manuscript presented in an intelligible fashion and written in standard English?

Reviewer #3: Yes

Reviewer #4: (No Response)

Reviewer #5: Yes

Reviewer #6: Yes

6. Review Comments to the Author

Reviewer #3: Thank you for incorporating the limitation of duty to warn and touching on ethics. I think this is a great paper and your work is essential as AI continues to advance in the field of technology. Great work.

Reviewer #4: (No Response)

Reviewer #5: This is a very interesting study in which the authors demonstrated the potential use of AI in mental health settings. I believe the paper is ready for publication. However, I have a few minor suggestions that could help improve the clarity of the manuscript:

1. Please ensure that all abbreviations are spelled out in full when first introduced. I know that some abbreviations are common, but some readers might not be familiar with them (e.g. IRB on page 10).

2. It might be helpful to provide some background in the Methods section about the first author. Since the first author created the content and prompts for the vignettes, there may be some subjective influence embedded in the process. Offering some background and reflexivity would add valuable context.

3. Personally, I would suggest moving the panel characteristics to the Results, as it aligns more with the study's findings than the Methods. However, this decision is ultimately up to the authors.

Overall, I believe the study is ready for publication.

Reviewer #6: Reviewer comments

I was keen to read this paper from the title and it did not disappoint! This is an engaging exploration of a topical issue, and it was great to see all the data and materials on osf. The focus on linguistic patterns was also novel and worthwhile. The quality of writing was high and the use of Bayesian statistics very welcome.

I can see the amount of work that’s gone into responding to the first round of reviewer comments and the paper has definitely improved as a result. My main concern with the paper as it stands is the overwhelmingly optimistic outlook on the use of ChatGPT for couples therapy with limited consideration of the potential pitfalls. To some extent this might be a difference in opinion, and the authors word their claims carefully, however, I find the authors claim that: “GenAI has the powerful potential to meaningfully and linguistically compete with mental health experts in couple-therapy-like settings” a little misleading. Furthermore, there is a scientific/ethical imperative to highlight some of the future issues and considerations for further development of therapy with LLMs. Therefore, some key issues which merit further attention are:

The quality and suitability of the expert group

• Almost a third of the professionals recruited have no expertise in the field of couples therapy. This seems odd given the premise of pitting ChatGPT against ‘experts’. The paper is careful not to say ‘couples therapists’ but the fact that the study consists of couples therapy vignettes indicates a better group of experts would have been practicing couple therapists (whether clinical psychologist/psychotherapists etc. seems less relevant although still pertinent information to include). Indeed, whether the experts are currently practicing, and for how many years they have practiced is also important information when we are assessing whether ChatGPT can compete with therapists.

• Furthermore, I also wanted to know how many of the experts had clinical experience vs. had published in this area as there is a large gap between theory and practice and goes back to my previous point around a suitable comparison group.

• “Originally, a group of 15 experts was recruited but 2 discontinued participation.” – how were the experts recruited?

• The authors do highlight several limitations with the study that could threaten the generalisability of the results. However, it is unclear why ChatGPT is being compared with “mental health experts in couple-therapy-like settings” at all. For example, a clinical psychologist primarily trained in CBT or an academic who has published on relationship therapy would not deliver couples’ therapy in the real world, so why are these people being compared to ChatGPT in a ‘couple-therapy-like’ setting? Given this context, the “powerful potential [for ChatGPT] to meaningfully and linguistically compete with mental health experts” feels much less impressive.

Ethical and practical issues with LLMs as therapists

• There is a brief nod to possible harms in the Implications section but considering the scope of harm, the untested nature of LLM-guided therapy and importance of therapeutic work, this needs more engagement.

• The main implication is framed in terms of creativity vs. cost but this seems to downplay the potentially life-changing ramifications of ‘getting it wrong’ in a therapeutic context. What if ChatGPT does not intervene at the right moment between suicidal ideation to intent? What if ChatGPT advised a couple to split up? What about legal liabilities? Data storage and confidentiality? There have already been high-profile data breaches of therapeutic data (e.g. the Vaastamo data breach in Finland), although these were the transcripts of human therapists and their clients. There is also no guarantee that the long-term outcomes of couples therapy delivered by a chatbot would be better or comparable to a couples therapist e.g. the text format could be less effective or the lack of validation/acceptance by another person could affect outcomes.

• “Indeed, responsible therapists and practitioners need to closely supervise integrating GenAI into therapeutic processes.” – so there is a key role for human therapists to play. Could you explain your reasoning further? As this conclusion comes as somewhat of a surprise after the rest of the article.

• “mental health experts are desperately needed to ensure these models are being carefully trained and supervised to ensure the highest quality of care.” – if GenAI is to be used in therapy, then I absolutely agree. As this is the parting shot of this paper, I would expect more build up and expansion of this point, which seems to have been only briefly dealt with in the discussion (see my previous comment). How would experts be involved? Do you mean supervised in a machine learning sense or in a clinical sense? This speaks to one of the core, and very important, applications of this paper, and it would be very beneficial to hone in on this more. The authors don’t need to make precise recommendations but some guidance in this area would be useful for future research/ethical and effective development of GenAI in therapy.

Comments for the remainder of the paper, in chronological order

I wasn’t able to access the actual responses from the couples therapists/ChatGPT through osf – is this something that will be made available after publication? It would be fascinating to read their actual responses and speaks to the core open science principle (and the journal’s requirements) to have all the data available.

“early and mounting evidence suggests that generative artificial intelligence (GenAI) may prove helpful in psychotherapy (1).” – this is not a suitable reference, as it is the Turing reference from 1950 so does not speak to increasing evidence for use in therapy.

“For example, HAILEY has been proposed as an addition to TalkLife” - Please define what ‘HAILEY’ is.

“These innovations have been met with great success compared with previously federally-funded efforts (13, 15, 17, 18). Indeed, recent trials of the ePREP and OurRelationship (two evidence-based) programs have served relationally distressed ethnic, racial, and sexual minority couples, demonstrating small to medium-sized effects (13, 19, 20).” – I was interested here to know what the effect sizes were from the federally-funded programmes, since the outcomes from the digital programmes were portrayed as being more successful?

“Finally, given the group of experts had several possible responses to select the best response, ChatGPT was given the same opportunity and was rated by the research team (i.e., authors 1-6 and 18)” – did the research team have any expertise in couples therapy which facilitated this selection? I am assuming so but think it would be clearer to explicitly state as such, and what the level of expertise was.

“Estimates aggregated across all vignettes revealed responses written by ChatGPT (μ = 27.72, σ = 0.83) were rated higher on the common factors of therapy than those written by therapists (μ = 26.12, σ = 0.82), indicating a large and reliable difference (d = 1.63, 95% CI [1.49, 1.78]) favoring ChatGPT (see Table 3).” – Could these aggregates and CIs be included in Table 3? As it is a little confusing to be referred to a table for figures that do not directly appear there.

“It appeared that controlling for response length and part-of-speech reduced most of the effect sizes, but made all of the estimates less precise.” – so this implies that if the therapists had been asked to create longer responses, their responses would have been rated more highly? I wonder if the size of the text boxes on the survey had any influence on this? E.g. if a text box is a medium size, then a person would intuitively provide a medium-length answer as it looks as though that is what is ‘required’ in this instance whereas ChatGPT would have no such psychological processes at work.

“The second post hoc aim indicated that when the panel of participants perceived a response to be written by ChatGPT, they rated the response lower. This is especially interesting given that participants appeared to favor the ChatGPT’s responses to the vignettes over the therapist’s responses and may be indicative of an underlying technophobia towards ChatGPT behaving as a therapist.” – or rather than technophobia, could be an indication of a rational response to interacting – and building a relationship with – a machine vs. a person? When we know from decades of research it is the relationship that heals.

The conclusion is very engaging and well-written but is quite abstract and works as more of an overview of the field than a conclusion to this study. It would benefit from being slightly more focussed on the current study.

7. PLOS authors have the option to publish the peer review history of their article (what does this mean?). If published, this will include your full peer review and any attached files.

**Do you want your identity to be public for this peer review?** For information about this choice, including consent withdrawal, please see our Privacy Policy.

Reviewer #3: No

Reviewer #4: No

Reviewer #5: **Yes: **Duaa H. Alrashdi

Reviewer #6: No

---

## [Editor Report · Decision Letter 2]

10 Dec 2024

When ELIZA Meets Therapists: A Turing Test for the Heart and Mind

PMEN-D-24-00239R2

Dear Mr. Hatch,

We are pleased to inform you that your manuscript 'When ELIZA Meets Therapists: A Turing Test for the Heart and Mind' has been provisionally accepted for publication in PLOS Mental Health.

Best regards,

Brian Bauer, PhD

Academic Editor

PLOS Mental Health